# Role of Non-Thermal Plasma in *Fusarium* Inactivation and Mycotoxin Decontamination

**DOI:** 10.3390/plants12030627

**Published:** 2023-01-31

**Authors:** Pratik Doshi, Božena Šerá

**Affiliations:** Department of Environmental Ecology and Landscape Management, Faculty of Natural Sciences, Comenius University Bratislava, Ilkovičova 6, 84215 Bratislava, Slovakia

**Keywords:** cold plasma, *Fusarium*, mycotoxin

## Abstract

*Fusarium* spp. is a well-studied pathogen with the potential to infect cereals and reduce the yield to maximum if left unchecked. For decades, different control treatments have been tested against different *Fusarium* spp. and for reducing the mycotoxins they produce and are well documented. Some treatments also involved integrated pest management (IPM) strategies against *Fusarium* spp. control and mycotoxin degradation produced by them. In this review article, we compiled different control strategies against different *Fusarium* spp. In addition, special focus is given to the non-thermal plasma (NTP) technique used against *Fusarium* spp. inactivation. In a separate group, we compiled the literature about the use of NTP in the decontamination of mycotoxins produced by *Fusarium* spp., and highlighted the possible mechanisms of mycotoxin degradation by NTP. In this review, we concluded that although NTP is an effective treatment, it is a nice area and needs further research. The possibility of a prospective novel IPM strategy against *Fusarium* spp. is also proposed.

## 1. Introduction

One of the most important plant-pathogenic groups is *Fusarium* spp., causing different diseases in agricultural crops. Several devastating diseases such as seedling blight, root rot, and Fusarium crown rot are caused by *Fusarium* spp. [1]. Different species of *Fusarium* genera, such as *F*. *graminearum*, *F. culmorum*, *F. poae*, and *F. avenaceum* are economically important in Europe [1]. Disease development and mycotoxin production can be influenced by more than one *Fusarium* spp., which are often found to interact with each other [1,2,3].

Summerell [4] accessed the American Phytopathology Society website only to find that 83 out of 108 plant species, from the list of diseases on agricultural and horticultural crops, have more than one *Fusarium* disease impacting their production. One of the most important diseases is the Fusarium head blight (FHB) caused predominantly by *Fusarium graminearum*. Fusarium head blight is a global concern due to its severe impact on grain quality and yield [5], which still needs considerable attention for its devastating effects. Buerstmayr et al. [5] stated in their review that an estimated loss of approximately USD 2.5 billion was attributable to FHB on wheat and barley for the period 1993–2001.

This genus of pathogenic fungi is known not only in agriculture but also in the food industry. There are toxic compounds produced by certain fungi called mycotoxins. The mycotoxins produced by *Fusarium* spp. that are most studied majorly belong to Trichothecenes (Deoxynivalenol (DON), T-2 and HT-2), Zearalenone (ZEN), and Fumonisins [6,7]. In Europe, Fumagalli [8] stated in their review that the World Mycotoxin Survey, published in 2020, showed that mycotoxin risk was found to be high to severe compared to previous years, with deoxynivalenol (DON) found to be the biggest threat. *Fusarium* spp. produces deoxynivalenol mycotoxin (predominantly by *F. graminearum* and *F. culmorum*), which infects several cereals and can cause yield losses of up to 50% [9]. It has been found that DON mycotoxin can cause vomiting, digestive disorders, oxidative damage, and reproductive toxicities in animals and humans when ingested [7]. Deoxynivalenol also has a toxicological impact on poultry and animal health [10]. Co-contamination of these mycotoxins is also on the rise, while work on identifying new compounds is still ongoing [7]. Due to these reasons, the World Health Organization (WHO) and Food and Agriculture Organization (FAO) have declared a high-priority on mycotoxins [8]. Realizing both that *Fusarium* spp. is a threat to the agriculture sector and that the toxic effects of mycotoxins are produced by these fungi, there are different ongoing control measures, and some researched in the past against both inactivations of *Fusarium* spp. and mycotoxin decontamination. However, concerning chemical control, unfortunately, *Fusarium* spp. is becoming resistant to some chemical fungicides due to their indiscriminate and long-term use [3]. Therefore, it is essential to search for novel, non-chemical control strategies against *Fusarium* spp. inactivation and mycotoxin decontamination.

One such physical control method, known as the non-thermal plasma/cold plasma (used interchangeably) technique, has started showing promising results in both *Fusarium* spp. inactivation and mycotoxin decontamination. Physical plasma is the fourth state of matter. It is a partially or fully ionized quasi-neutral substance that is made up of electrons, ions, neutral particles, molecules in the ground or excited state, radical species, and quanta of electromagnetic radiation (UV photons and visible light) [11,12]. These particles exhibit collective behavior. Since this review does not include thermal plasma, only non-thermal plasma (NTP) and its applications are discussed hereafter.

A partially ionized gas with electron temperatures higher than ion temperatures will produce NTP [11]. Non-thermal plasma is gaining attention in various fields due to its ability to generate radicals and reactive species [13]. There are different types of NTP-generating devices, some of which are presented in Figure 1. The descriptions of the NTP devices are taken and explained from Domonkos et al. [12].

All the NTP devices in Figure 1 use working gas such as air, oxygen, nitrogen, inert gases (Argon, Helium), and their mixtures. The first is the Dielectric Discharge Barrier which comprises two electrodes, one of which is covered with dielectric material (such as quartz, glass, ceramics, enamel, silicon, rubber, teflon, mica, or plastic) and separated by an insulating dielectric barrier. Voltage is supplied to these electrodes to generate plasma. The sample is kept between the two electrodes.

The second is Plasma Jet. It consists of two concentric cylindrical electrodes. The inner electrode is connected to the power supply, which causes the ionization of the working gas. It produces a high-velocity stream of highly reactive chemical species with weak emitted light and shoots down in the form of a jet onto the sample.

In corona discharge, as the name suggests, the ionized gas creates a crown around the active electrode. Two or more electrodes are attached to the high voltage, and the coronizing electrode is generally in the form of a needle or thin wire.

Gliding arc discharge produces cold plasma, but only under specific conditions. The electrodes are placed in a fast gas flow, which, when supplied with high voltage, discharges the voltages with increased volume and length in the flow direction. The plasma can be applied directly or indirectly on to the target object.

NTP has already shown promising results in the fields of medicine [14,15,16,17,18], in the textile industry [19,20,21], in forestry [22,23], in the food industry [24,25,26], and in the food preservation industry [27,28,29,30]. With so much knowledge about NTP and its established diverse applications, the focus has also shifted towards the agriculture sector (known as ‘plasma agriculture’ [31]) in recent years, as there are calls for new and clean technologies to replace chemical pesticides that are growing around the world. Several research articles have demonstrated that NTP can potentially improve seed germination and crop yield [32,33,34,35,36]. Although it is evident that NTP undoubtedly works well in the enhancement of different crop traits, we shift our focus onto the main topic of this review which is the effective use of NTP in plant protection, especially against *Fusarium* spp.

*Fusarium* spp. is one of the infamous plant pathogens that ranks in the list of the top 10 fungal pathogens in molecular plant pathology (Dean et al. [37]) and their derived mycotoxins. Therefore, in this article, (a) the application of NTP against *Fusarium* spp. inactivation was explored and reviewed, and (b) the decontamination/degradation of mycotoxins produced by *Fusarium* spp. was also reviewed. The possible underlying mechanisms for both were highlighted and discussed. Given that NTP is a fairly new technique in the agriculture sector, we also propose a combination of NTP and biological control as a possible IPM strategy that may offer better protection against *Fusarium* spp. and other pathogens, only after conducting extensive research to test their synergy.

## 2. Results

### 2.1. Effect of NTP/Cold Plasma on Inactivation of Fusarium *spp.* (Group 1)

Below in Table 1, we list the articles displaying the success and failures of different plasma techniques against the inactivation of *Fusarium* spp. spores and mycelium.

Homa et al. [38] explained the importance of sweet basil as a culinary herb. In addition, they discussed two different diseases, namely basil downy mildew (BDM) caused by *Peronospora belbahrii* and *Fusarium* wilt of basil (FOB) caused by *Fusarium oxysporum* f. sp. *basilici*, which causes significant damage to this crop. In search of new options to control the FOB disease, they tested cold plasma treatment against FOB mycelium on sweet basil plants and studied seed treatment. In the laboratory test, FOB plugs were treated for a period of 5, 10, and 15 min, while non-treated FOB plugs were used as the control. They found no significant difference in mean mycelial growth when cold plasma jet treatment on mycelium was performed. In the plants experiment, the six-leaf staged plants were inoculated with FOB 1 day before or after the cold plasma jet treatment. They found that prior treatment with cold plasma jet resulted in less stunting of plants. For the seed treatment, the seeds were inoculated with talc/chlamydospore mixture, followed by cold plasma dielectric barrier discharge treatment for 1, 5, 10, or 15 min. They found that as the time increased, the colonies on the seeds decreased, with the maximum reduction seen at 10 and 15 min post-treatment. Subsequent papers also concerned *Fusarium oxysporum,* where similar results were obtained by Panngom et al. [39] when they tested the effect of non-thermal plasma against *F. oxysporum* f.sp. *lycopersici* in a susceptible tomato variety. They not only found that the NTP germination of spores decreased over time after exposure to argon (Ar) plasma for 10 min, but also increased the transcription of pathogenesis related genes, concluding that NTP can also be used to up-regulate resistance mechanisms. This was also confirmed in another study conducted by Go et al. [40], where they found complete inhibition of mycelial growth, spore germination, and up-regulation of membrane-related gene (*SHO1*). The disinfection of seeds is not limited to cereals or crops but is also used in forestry, as demonstrated by Swiecimska et al. [22] when they treated scots pine seeds against *F. oxysporum* with non-thermal plasma for different periods viz. 1 s, 3 s, 5 s, 10 s, 15 s, 20 s, 30 s, and 60 s. They found that 3 s of treatment was the optimal time to disinfect the seeds.

Different plasma techniques were tested on inactivating *F. culmorum* or F. *nivale*. Zahoranová et al. [41] tested the effect of cold atmospheric pressure plasma on different fungal species, namely *Fusarium nivale, F. culmorum, Trichothecium roseum, Aspergillus flavus, A. clavatus,* which were used to contaminate wheat seeds artificially. They found that the efficacy of cold atmospheric pressure plasma was greatest on *F. nivale* and the least on *A. clavatus*. Zahoranová et al. [42] also investigated the effect of cold atmospheric pressure plasma on native microbiota and three pathogens (*Aspergillus flavus, Alternaria alternata* and *F. culmorum*) on the maize surface, and also on germination and growth parameters. They found that after 60 s of treatment, the native microbiota was completely devitalized and *F. culmorum* was reduced, while *A. flavus* and *A. alternata* were reduced after 300 s treatment. The study by Hoppanová et al. [43] also supports the argument that low-temperature plasma inhibited the growth of *F. culmorum* on the seed surface and, when combined with fungicide, complete inhibition of the fungus was achieved.

Bousba et al. [44] tested a mixture of different plasma working gas (He + air, He + O_2,_ and He + N_2_O) for the decontamination of fungus in polluted water. They found that different plasma gas mixtures prevented the growth of *Fusarium pseudograminearum* at different time intervals, and that the combination of He + N_2_O took the least time (5 min). In cereals, Wang et al. [45] effectively inactivated four major *Fusarium graminearum* strains (2 to 6 log_10_ reduction) under in vitro and in vivo conditions using cold atmospheric plasma. Under in vivo conditions, too, cold atmospheric plasma reduced the pathogenicity of *F. graminearum*.

Similar results were obtained by Chang et al. [46] where they tested a different plasma treatment, namely corona discharge air plasma (CDAP), which consists of nitric oxide and nitric dioxide, on the decontaminating microbes on onion. They found that the isolation frequency of *Fusarium* spp. was less as compared to *Alternaria* spp. and *Botrytis* spp. Additionally, different treatment concentrations showed different efficiencies. In the case of *Alternaria* spp., 2~2.6 ppm of O_3_ slightly stimulated the mycelial growth, whereas 20~24 ppm of O_3_ gradually inhibited the fungus. *Botrytis* spp., on the other hand, showed different time-dependent results. Slight inhibition was seen in *Botrytis* spp. with a treatment of less than 4 h, while the 8 h of treatment promoted growth, irrespective of O_3_ concentration. The conidial germination of both the fungi viz. *Alternaria* and *Botrytis* spp. was strongly inhibited at 13.7~14.4 ppm of O_3_. Sandanuwan et al. [47] tested the cold plasma technique against different fungal pathogens of the Cavendish banana fruit. They successfully decreased the percentage disease index (PDI) of *Colletotrichum musae*, *Fusarium semitectum*, and *Colletotrichum gloeosporioides* using cold plasma compared to control and also fungicide treatments. Meanwhile, in pine seeds, Šerá et al. [23] demonstrated that pine seeds contaminated with *Fusarium circinatum* were completely disinfected using non-thermal plasma technique after 5, 10, 60, 180, and 300 s and the inoculated seeds remained microbe-free for 12 days after 60 s plasma treatment, indicating that seeds can be kept viable with this treatment.

### 2.2. Effect of Non-Thermal Plasma/Cold Plasma on Degradation/Decontamination of DON Mycotoxin (Group 2)

Below, in Table 2, we list the articles displaying the success of different plasma techniques used to decontaminate *Fusarium* spp. mycotoxin and mycotoxins from other plant pathogens.

Ten Bosch et al. [48] tested cold atmospheric pressure plasma on mycotoxin degradation by different microorganisms. The study selected DON, zearalenone, enniatins, fumonisin B1, and T2 toxin produced by *Fusarium* spp., sterigmatocystin produced by *Aspergillus* spp., and AAL toxin produced by *Alternaria alternata*. While they found that 60 s exposure to CAPP resulted in complete degradation of pure mycotoxin, sterigmatocystin offered the highest resistance. They concluded that CAPP is an efficient technique to degrade mycotoxin and the degradation rates may vary due to the mycotoxin structure. The argument is also supported by Abbasian et al. [49], as they succeeded in degrading DON mycotoxin from *Fusarium* spp. using Argon plasma jet, concluding that certain conditions, such as time and concentration of mycotoxin in food, are required for the plasma treatment to work efficiently.

Wang et al. [45] also investigated the effects of cold atmospheric plasma against DON mycotoxin and also found that the treatment inhibited DON biosynthesis in vitro. Guo et al. [50] tested the effect of cold plasma against DON and ochratoxin A (OTA) mycotoxins. They found that 8 min treatment with cold plasma significantly reduced both DON and OTA mycotoxin. Further they found that mycotoxin reduction was directly proportional to increase in cold plasma treatment time. Qiu et al. [51] investigated the effect of plasma-activated water (PAW) on wheat contaminated with DON mycotoxin. They found a significant decrease in the DON mycotoxin; the number of bacterial, fungal counts, and surviving *F. graminearum* in wheat was significantly reduced. Other mycotoxins such as T-2 and HT-2 from *Fusarium* spp. were also tested with the atmospheric cold plasma technique [52]. They artificially spiked the wheat grains with the mycotoxins and were treated with the cold atmospheric plasma for 10 min. The treatment reduced the pure T-2 by 63.63%, while HT-2 concentrations reduced by 51.5%, the spiked T-2 concentration was reduced by 79.8%, and the HT-2 by 70.4%.

## 3. Discussion

### 3.1. Effect of NTP/Cold Plasma on Inactivation of Fusarium *spp.* (Group 1)

The unique quality of NTP is the working gas used either individually or in combination with other gases, according to the study and apparatus. The studies reviewed, in the results, whether the plasma conditions were the same or different, and whether they had the same outcomes of inactivating *Fusarium* spp. and other fungi in their respective study.

It is evident that NTP has potential plant protection properties and is effective against *Fusarium* spp. and other studied fungi. For instance, Rüntzel et al. [53] reported that cold plasma treatment of 10–30 min effectively inactivated the fungi (*Aspergillus* spp. and *Penicillium* spp.) from the surface of black beans (*Phaseolus vulgaris* L.). Similar results were confirmed by using plasma-activated water on the inactivation of *Penicillium italicum* in kumquat [54]. Ahmad et al. [55] tested two plasma treatments viz plasma-activated water and plasma activated H_2_O_2_ solution on the spores and mycelium of *Colletotrichum gloeosporioides* which causes anthracnose disease in pepper (*Capsicum annuum* L.) seeds. They found that both treatments effectively inhibited the spore and mycelium of *C. gloeosporioides*. This result is in line with the study conducted [47]. Not only fungi but aerobic bacteria, yeasts, and molds were also inactivated using a corona discharge plasma jet, where 5 min of treatment saw 1.0 log reduction [56]. They also stated that their susceptibility to plasma treatment varied due to different structure and chemical composition of microbes.

It is also crucial to understand the different underlying mechanisms of fungal inactivation caused by NTP, irrespective of the treatment conditions and apparatus used in the respective studies. Several possible theories, such as from Go et al. [40], observed that fungal spores treated with the plasma showed severe structural changes and were crushed and shrunk. They stated that this change was due to the reaction with the active species formed during the plasma process (see [39,40,57]), and further explained in detail that membrane lipids of the microorganism are affected by the reactive species in plasma and that, additionally, the oxidation of amino acids and nucleic acids is detrimental. Supporting this argument, Wang et al. [45] stated that the mode of inactivation was due to the destruction of the cell membrane, accumulation of intracellular ROS, and depolarization of the mitochondrial membrane. Other studies also mentioned cytoplasmic leakage as one of the possible mechanisms of inactivating plant pathogens [58,59]. We hypothesize that *Fusarium* spp. spores may interact with the reactive species from the plasma gas, and disruption in the spores may lead to cytoplasmic leakage, which needs to be investigated for confirmation in the future.

Homa et al. [38] argue that pathogen deactivation depends on the number of factors, such as the host to be treated, the pathogen(s), the type of cold plasma system, and the degree of exposure of the cold plasma on the host organism. With respect to the cold plasma system, in our review, the most common gas supply was seen to be air or atmospheric plasma.

Scholtz et al. [31] mentioned that it is difficult to compare the results of Shaw et al. [60] and Khun et al. [61] as they use two different plasma treatment conditions (such as gas supply, voltage, current, power, frequency, etc.) on the same reference microbe. It is important to address that studies using similar or the same plasma treatment conditions, for instance, regarding this review, the studies conducted by Swiecimska et al. [22], Zahoranová et al. [41], Zahoranová et al. [42], and Hoppanová et al. [43] can easily be reproduced, compared, and concluded.

Additional biological factors mentioned by Adhikari et al. [62], such as the genus and species of the plant, the microenvironment of the plant–pathogen system, the species and strain of the pathogen, the structure of the cellular envelopes, and the microbial growth phase, could also possibly affect the pathogen deactivation and need to be carefully considered in the future studies to confirm if they have any influence on the outcome.

### 3.2. Effect of NTP/Cold Plasma on Degradation/Decontamination of DON Mycotoxin (Group 2)

From the results interpreted, NTP is found to successfully degrade mycotoxins, especially DON produced by *Fusarium* spp., and mycotoxins from other phytopathogens from different studies. Zhang et al. [63] subjected DON mycotoxin solution to optimized conditions of double dielectric barrier discharge, only to find degradation of 98.94% within 25 min of plasma treatment. Ott et al. [64] also confirmed the degradation of more than 99% of 100 μg DON mycotoxin in aqueous suspensions after 21 min of direct high voltage atmospheric cold plasma treatment, using air to generate reactive oxygen and reactive nitrogen species. They also reported a much lower degradation (33%) of 100 μg DON mycotoxin in powdered form. Janić Hajnal et al. [65] tested atmospheric cold plasma treatment against decontamination of *Alternaria* toxins (alternariol (AOH), alternariol monomethyl ether (AME), and tentoxin (TEN)) content in wheat flour. They artificially spiked the wheat flour with these toxins and subjected it to cold atmospheric plasma treatment. After 180 s and with treatment performed at 6 mm from the plasma source, the best results were obtained with reductions of 60.6%, 73.8%, and 54.5% for AOH, AME, and TEN, respectively. With such growing evidence, it can be safely said that NTP could have the edge over thermal treatment. To support this argument, Varilla et al. [66] argued that thermal treatment, such as cooking and pasteurization, cannot be a reliable solution for mycotoxin decontamination as some of the mycotoxins are resistant to thermal treatment.

Very few studies have been dedicated to understand the possible mechanisms of how non-thermal plasma degrade mycotoxins. One possible simple mechanism is that the degradation is caused by some energetic particles. For example, ten Bosch et al. [48] found only 60 s is enough for almost complete degradation of many studied mycotoxins. The temperature of the gas and substrate during NTP treatment is usually lower than 60 °C. At such temperatures, even sensitive proteins do not degrade, nevertheless, 60 s was quite sufficient for mycotoxin degradation. They further argued that the mycotoxin type and the matrix greatly influenced the inactivation efficacy of the plasma treatment. In relation to this argument, it is important to understand that different mycotoxins have different chemical structures and may undergo multistep degradation [52,67]. One of the possible reasons also stated by ten Bosch et al. [48] is the reactive species generated in the plasma. Under continuous voltage and due to chemical reactions in the plasma, reactive species such as O, O_3_, OH, NO_X,_ when they interact with the pure compounds of mycotoxins, lead to the fragmentation of molecular bonds, which further leads to the production of volatile compounds (Iqdiam et al. [52]) that are known to be less toxic [67]. This argument is further supported by Qiu et al. [51] using PAW, where they found that the DON degradation rate was directly proportional to an increase in exposure time, as there was an increase in the concentration of reactive species such as long-lived particles (H_2_O_2_, O_3_, H^+^, NO_2_^−^, NO_3_^−^) and short-lived particles (OH, O_2_^−^, NO, and ONOOH). Gavahian and Cullen [68] also proposed that several properties of plasma, such as the concentration of oxygen, hydroxyl radicals, the presence of photons, and ultraviolet radiation, could affect mycotoxin degradation using plasma treatment.

Another possibility is from the study of Wang et al. [45]. They found that mycotoxin degradation using cold atmospheric plasma is achieved through reduced acetyl-CoA production, toxisome formation, and key trichothecene biosynthetic gene (TRI) expression, and in vivo by inactivation of fungal spores, thereby reducing DON production.

The mechanism of mycotoxin degradation using NTP in vivo, as proposed by Gavahian and Cullen [68], can be attributed to the principle of “killing two birds with one stone”. They proposed that the reactive species produced by plasma alters the cell membrane and cell walls to release cytoplasm leading to cell inactivation, which does not allow the fungi to produce mycotoxin. Additionally, the plasma reacts with the fungal cell on multiple sites, resulting in loss of functions and eventually apoptosis [69], which Lee et al. [70] support in their study while investigating the effect of plasma on the spores of *Cordyceps bassiana*.

There have been relatively few articles on the selected topic which are useful. NTP is a very progressive perspective direction in agriculture and plant protection when it comes to finding chemical-free alternatives. Nevertheless, NTP has several limitations. For instance, Šimončicová et al. [71] mentioned that high investment cost, maintenance, and servicing cost are limiting the use of cold atmospheric plasma. Further, up-scaling NTP to the industrial level for decontamination is still far from realization [68].

Another challenge with NTP is the treatment of the uneven surface. Gavahian and Cullen [68] argued that NTP being a surface treatment may not be effective against irregularly shaped or bulky food material. There is a high chance that fungal spores or mycelium may not get treated in irregularly-shaped seeds when treated with NTP, leaving them in the infective state.

There have been some reports that state that NTP has a potential negative impact on food lipids [68]. The studies of Varilla et al. [66] and Jadhav and Annapure [72] confirmed the above argument in their respective studies, as they found that NTP induces lipid oxidation in the meat tissues and fish, thereby turning them inedible. Apart from lipids, oligosaccharides found in juices are also degraded by ozonolysis, triggered by cold plasma treatment [73].

Many research shows no or minimal impact on physical and chemical attributes of some food products. Possible negative effects could be found in other food products entering the human food chain. Such food products need further attention, and research to optimize NTP conditions may be needed. This means there need to be specific guidelines for using NTP treatment on specific food products to enhance their quality. In this way, a compilation of food products with similar or the same treatment can be performed, which can help to upscale the results to an industrial level in the future.

In agriculture, especially in the case of microbial inactivation and/or mycotoxin degradation, perhaps, in the future, a combined strategy of NTP and biological control could be tested as a part of the Integrated Pest Management strategy. We recommend investigating the effects of NTP against different biological control agents/biocontrol agents under laboratory and greenhouse conditions. This will help us to identify if there is a synergistic or antagonistic relationship between the two treatments. Different gases can also be tested to understand the direct effect of gases on these biocontrol agents. These results can be further tested under greenhouse conditions to understand their interaction and efficacies before testing them in the field trials.

## 4. Materials and Methods

The methodology of the work is based on the analysis and subsequent synthesis of literary sources. We searched the literature available on the Web of Science Core collection database. For convenience, we divided the methodology into two groups viz. Effect of NTP/Cold plasma on inactivation of *Fusarium* spp. (Group 1) and Effect of NTP/cold plasma on degradation/decontamination of DON mycotoxin (Group 2).

For Group 1, we searched using the terms ‘cold plasma’ and ‘*Fusarium*’. The search displayed a total of 30 articles, and the results gave both inactivation of spores and mycelium of *Fusarium* spp. and other fungal species. We filtered the search by excluding the review articles, as we wanted to use them for introduction and discussion, and by excluding the other fungi except for *Fusarium* spp. We also excluded the results that displayed mycotoxin. Eventually, we selected 12 articles and used them for the systematic review. The articles were selected and reviewed after following PRISMA statement guidelines [74] (Figure 2).

In the case of Group 2, Effect of NTP/cold plasma on degradation/decontamination of DON mycotoxin, we searched the Web of Science core collection database by using the terms ‘cold plasma’ AND ‘Fusarium’ AND ‘deoxynivalenol’. The search returned with nine articles of which we excluded the three review articles. The remaining six articles were considered and used for the systematic review. The articles were selected and reviewed following PRISMA statement guidelines (Page et al., 2021) [74] (Figure 3).

## 5. Conclusions

NTP is undoubtedly one of the most innovative physical control agents of plant pathogens and mycotoxin decontamination. This article attempts to provide a focused view on the use of NTP against the inactivation of *Fusarium* spp. and other studied plant pathogens, and the degradation of mycotoxins produced by *Fusarium* spp. and other fungal species. The presented overview shows that rather than the apparatus per se, the plasma treatment conditions such as gas used, voltage, power, and treatment time are the most important conditions that can be used for comparative analysis to study its efficiency. Although there are several possible explanations of the mechanisms of fungal inactivation and mycotoxin degradation, it is still a niche area that needs thorough research and understanding. Additionally, it will be interesting to test NTP and biological control under the Integrated Pest Management framework to find if there is a possibility to achieve synergy that could replace the traditional practices of chemical treatment, and ensure sustainable agriculture in the future.

## Figures and Tables

**Figure 1 plants-12-00627-f001:**
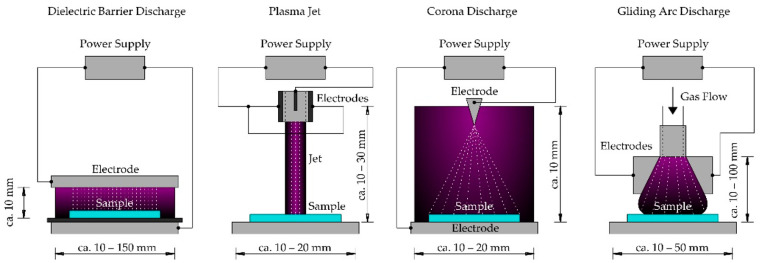
Types of different non-thermal plasma generating devices (adapted from Domonkos et al. [12]).

**Figure 2 plants-12-00627-f002:**
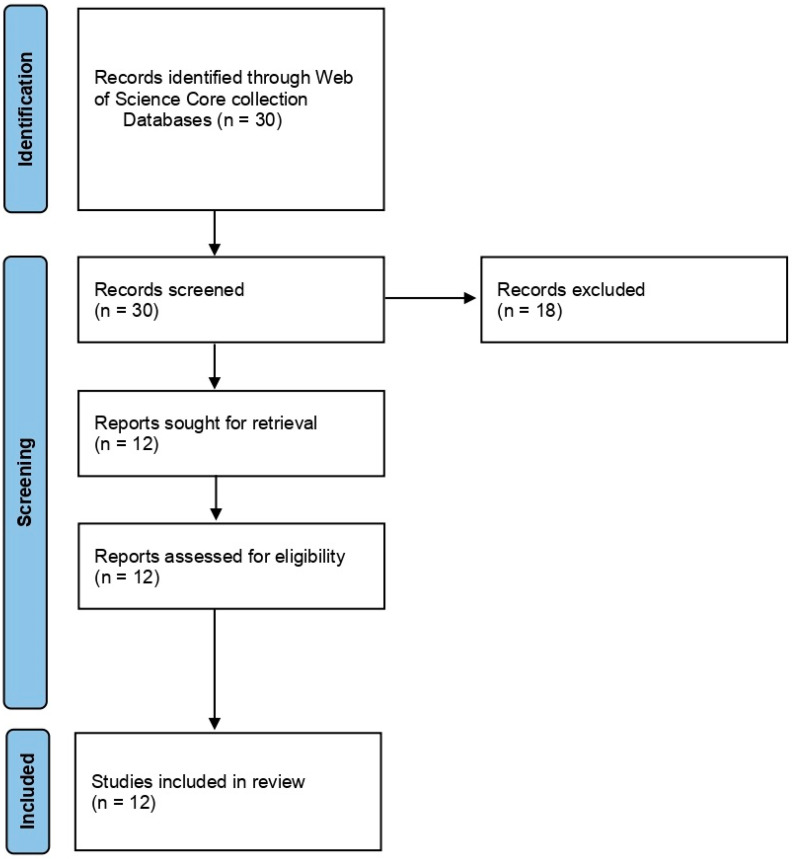
PRISMA flowchart summarizing the information gathering sequence and selection for the systematic review of Group 1: Effect of NTP/Cold plasma on inactivation of *Fusarium* spp.

**Figure 3 plants-12-00627-f003:**
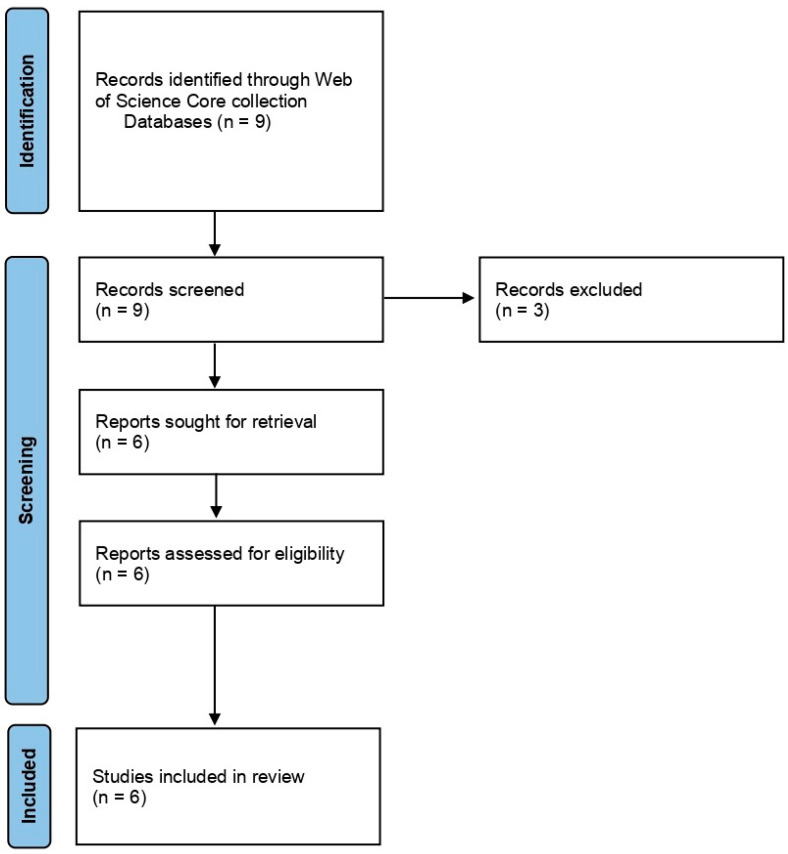
PRISMA flowchart summarizing the sequence of information gathering and selection for the systematic review of Group 2: Effect of NTP/cold plasma on degradation/decontamination of DON mycotoxin.

**Table 1 plants-12-00627-t001:** Summary of the research on “Non-thermal plasma”/“cold plasma” against inactivation of *Fusarium* spp. spores or mycelium.

*Fusarium* spp.	Pathogen(s) Under Study	Test Crop/Plant/Tree/Subject	Plasma Treatment Type	Plasma Treatment Conditions	Salient Results	References
*Fusarium oxysporum*	*Fusarium oxysporum* f. sp. *basilici*	Sweet basil	Cold plasma	G: He V: 13 kV P: 15 W Fr: 28.8 kHz Time: 5, 10, or 15 min	Cold plasma jet showed no significant effect on mycelium. Direct cold plasma jet on seedlings and cold plasma dielectric barrier discharge on seeds exhibited varying efficacies.	[38]
	*Fusarium oxysporum* f.sp. *lycopersici*	Tomato	Non-thermal plasma treatment	G: air or argon V: 0.75 kV P: 7.5 W C: 80 mA	Fungal spore germination was reduced over time after exposure to plasma treatment for 10 min. Necrotic death was observed in the majority of treated spores. Increased transcription of pathogenesis-related genes.	[39]
	*Fusarium oxysporum*	Paprika in spices	Non-thermal atmospheric plasma (NTAP)	G: air P: 1000 W Fr: 28 kHz Time: 0, 10, 30, 45, and 90 s	NTAP showed complete inhibition of *F. oxysprum* spores and mycelial growth. Loss of plasma membrane and up-regulation of membrane-related gene (*SHO1*). In vivo, 50% inhibition of fungal pathogens after 90 s treatment.	[40]
	*Fusarium oxysporum*	Pine seeds	Non-thermal plasma	G: air V: 20 kV P: 400 W Fr: 14 kHz Time: 1, 3, 5, 10, 15, 20, 30, and 60 s	3 s of plasma treatment was optimal for inhibiting *F. oxysporum* growth.	[22]
*Fusarium culmorum*	*Fusarium nivale, F. culmorum, Trichothecium* roseum, Aspergillus flavus and Aspergillus clavatus	Wheat	Cold Atmospheric Pressure Plasma	G: air V: 20 kV P: 400 W Fr: 14 kHz Time: 30–300 s	The efficiency of plasma treatment decreased in the following order: *Fusarium nivale* > *F. culmorum* > *Trichothecium roseum* > *Aspergillus flavus* > *A. clavatus*.	[41]
	*Aspergillus flavus, Alternaria alternata* and *Fusarium culmorum*	Maize	Cold Atmospheric Pressure Plasma	G: air V: 20 kV P: 400 W Fr: 14 kHz Time: 30–300 s	F. culmorum was reduced by 3.79 log (CFU/g) after 60 s treatment, while the reduction in *A. flavus* and *A. alternata* was found by 4.21 log (CFU/g) and 3.22 log (CFU/g), respectively, after 300 s plasma treatment.	[42]
	*Fusarium culmorum*	Wheat and Barley	Low-temperature plasma (LTP)	G: air V: 20 kV P: 400 W Fr: 14 kHz Time: 15, 30, and 60 s	Plasma treatment of 120–300 s significantly inhibited *F. culmorum* on the seed surface. A combination of plasma and chemical fungicide proved more effective.	[43]
*Fusarium graminearum*	*Fusarium pseudograminaerum*	Fungal polluted water	Plasma working gas mixtures	G: He, O_2_, N_2_O V: 14 kV Fr: 5 kHz Time: 5–25 min	He + N_2_O and He + Air treatment for 5 and 25 min, respectively, decontaminated the water; whereas He + O_2_ had the opposite effect and allowed fungal growth after treatment.	[44]
	*F. graminearum* HX01, *F. graminearum* LY26, *F. pseudograminearum* and *F. moniliforme*	Wheat	Cold atmospheric plasma	G: air V: 2 kV P: 5 ± 0.15 W Fr: 7 kHz Time: 0, 1, 2, and 3 min	In vitro, cold atmospheric plasma effectively inactivated all strains of the fungi. In vivo, cold atmospheric plasma inactivated fungal spores.	[45]
*Fusarium* spp., *Alternaria* spp. and *Botrytis cinerea*	*Fusarium* spp., *Alternaria* spp. and *Botrytis cinerea*	Onion	Corona Discharge Air Plasma (CDAP)	G: Ozone V: 20 kV Fr: 60 Hz Time: 6 h/day	A low concentration of O_3_ stimulated growth, and a high concentration inhibited growth in *Alternaria* spp. *Botrytis cinerea* showed time-dependent results: with lower time, growth was inhibited, and with higher time treatment, growth was promoted.	[46]
*Colletotrichum musae, Fusarium semitectum*, and *Colletotrichum gloeosporioides*	*Colletotrichum musae, Fusarium semitectum*, and *Colletotrichum gloeosporioides*	Cavendish banana	Cold plasma	G: air V: 15 kV Time: 0, 0.5, 1, 2, 3 min	The percentage disease index (PDI) in cold plasma was significantly lowered.	[47]
*Fusarium circinatum*	*Fusarium circinatum*	Pine seeds	Non-thermal plasma treatment	G: air V: 10 kV P: 400 W Fr: 14 kHz Time: 30–300 s	Reduction of seed-borne pathogens by 14–100%. Inoculated seeds remained mold-free for 12 days post-plasma treatment of 60 s.	[23]

G = Gas used, V = Voltage, P = Power, Fr = Frequency, C = Current.

**Table 2 plants-12-00627-t002:** List of the conducted research on “non-thermal plasma”/“cold plasma” in degradation/decontamination of Deoxynivalenol (DON) mycotoxin produced by *Fusarium* spp.

Pathogen(s)	Mycotoxin(s) Studied	Plasma Treatment Type	Plasma Treatment Conditions	Salient Results	References
*Fusarium*, *Aspergillus* and *Alternata* species	AAL toxin, enniatin A, enniatin B, fumonisin B1, sterigmatocystin, deoxynivalenol, T2-toxin, and Zearalenone	Cold atmospheric pressure plasma	G: air V: 38 kV P: 4 W/cm^2^ Fr: 17 kHz Time: 0, 5, 10, 20, 30, 60 s	All pure mycotoxins decayed after 60 s post-plasma treatment. Degradation rates varied due to mycotoxin structure and the matrix.	[48]
*Fusarium* spp.	Deoxynivalenol	Plasma jet	G: Argon Fr: 25 kHz Time: 60 s	Argon plasma jet destroyed both mycotoxin and *Fusarium* spp., producing a mycotoxin.	[49]
*F. graminearum* HX01, *F. graminearum* LY26, *F. pseudograminearum* and *F. moniliforme*	Deoxynivalenol	Cold atmospheric plasma	G: air V: 2 kV P: 5 ± 0.15 W Fr: 7 kHz Time: 0, 1, 2, and 3 min	Cold atmospheric plasma reduced DON production in wheat grains under in vivo conditions.	[45]
*Aspergillus niger, Rhizopus oryzae, Penicillium verrucosum, Fusarium graminearum*	Deoxynivalenol and Dochratoxin A	Cold plasma	G: air V: 25 kV Time: 2, 4, 6, and 8 min	Microbial activities were inhibited by cold plasma. DON and OTA mycotoxins were reduced by 61.25% and 55.64%, respectively.	[50]
*Fusarium graminearum*	Deoxynivalenol	Plasma activated water (PAW)	G: H_2_O_2_ and O_3_ V: 20, 30, 40, and 50 kV Time: 2, 4, 6, 8, and 10 min	DON mycotoxin reduced by 58.78% using PAW, and H_2_O_2_ and O_3_ were contributors to PAW.	[51]
T-2 and HT-2 standard toxins	T-2 and HT-2	Atmospheric cold plasma	G: air V: 0 to 34 kV P: 300 W Fr: 3500 Hz Time: 0, 2.5, 5, 7.5, and 10 min	Pure T-2 and HT-2 significantly reduced by 63.63% and 51.5%, respectively. After 10 min post-plasma treatment, mycotoxin spiked wheat grains reduced T-2 and HT-2 by 79.8% and 70.4%, respectively.	[52]

G = Gas used, V = Voltage, P = Power, Fr = Frequency, C = Current.

## Data Availability

Not applicable.

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
