# Peer review of "Role of Non-Thermal Plasma in *Fusarium* Inactivation and Mycotoxin Decontamination"

_plants, 2023, doi:10.3390/plants12030627_

Round 1
Reviewer 1 Report
This review article investigated the role of non-thermal plasma in Fusarium inactivation and mycotoxin decontamination. This should be an important issue to discuss since inactivation of mycotoxin decontamination is an essential issue for agricultural crops. Major suggestions are as follows,
1. It is suggested to add more discussion regarding application of Non-thermal plasma for agricultural crops. Are there any adverse outcomes for plants when applying this technique?
2. In abstract, it is suggested that major findings or information should be mentioned to highlight the importance or usefulness of this technology.
Author Response
Dear Reviewer,
We thank you for your patience and for providing critical review for our manuscript to make it better. Please find the answers (in italics) to your comments/suggestions below:
This review article investigated the role of non-thermal plasma in Fusarium inactivation and mycotoxin decontamination. This should be an important issue to discuss since inactivation of mycotoxin decontamination is an essential issue for agricultural crops. Major suggestions are as follows,
Thank you very much for your reviews, comments and suggestions.
- It is suggested to add more discussion regarding application of Non-thermal plasma for agricultural crops. Are there any adverse outcomes for plants when applying this technique?
Discussion is added in the text for other agricultural crops. Advantages and disadvantages of NTP treatment is also inserted into the text. Please refer to lines 22-239.
- In abstract, it is suggested that major findings or information should be mentioned to highlight the importance or usefulness of this technology.
The abstract has been modified as per your comments.
Reviewer 2 Report
Overall:
This review by Drs Doshi and Šerá investigates previous literature on the effect of Non-thermal plasma and it’s potential usage in inactivation of Fusarium species for Agriculture and mycotoxin degradation for the food industry. The use of NTP in agriculture poses an interesting replacement for pesticides and the aim of the review article appears to address it’s uses and previous findings associated with experimentation on Fusarium and a handful of other fungal species.
I find the submitted review article to be lacking in its ability to bring together a cohesive overview of the technology and previous findings in the literature. The manner in which the literature is presented is superficial and presented in a way that does not emphasise how the various findings are linked and is instead presented in an almost dot-point like fashion, ignoring any meaningful attempt at reviewing the topic as a whole. While it appears the authors have formatted their manuscript to be a systematic review, the PRISMA guidelines have not been followed.
The main articles reviewed are found in two tables in the submitted manuscript and the text associated with these tables addresses each of these articles one after the other, with little attempt at finding patterns within the literature and linking them together. I understand that the authors have tried to write a systematic review, however, this could be much better presented as a series of subjects important in the review (elaborated later on in my comments) where the authors discuss these subjects and the associated literature, rather than just listing papers. Good examples of systematic review structure can be found in https://www.mdpi.com/2223-7747/12/2/305 and https://www.mdpi.com/2223-7747/12/2/261
I also find detail lacking in the abstract and introduction, where thoughts appear to be unfinished and the reader is often left to their own devices to figure out what the author is trying to say.
In addition, the entire document needs to be edited for grammar.
I have included examples to all these concerns below. The examples however are not exhaustive as a rework of the entire document is required.
Comments per section:
Abstract:
The abstract gives no indication to the reader about the outcomes of the review. Does NTP work to inactivate Fusarium mycotoxin? In fact, mycotoxins are not mentioned at all in the abstract. This abstract is too superficial and short to give the reader any indication about what the review of the literature found. I found myself unsure about what the authors were trying to review based on the abstract alone. For a systematic review, be sure to follow the PRISMA for Abstracts checklist.
Introduction:
The introduction comes across very shallow and needs to be improved upon. Many points are raised and not elaborated on, examples include “one of the most important is the Fusarium head blight”, “The diseases caused by Fusarium spp. often involves more than one species in a form of a complex” and “World Health Organization (WHO) and Food and Agriculture Organization (FAO) has declared a high priority on mycotoxins”.
Why is Fusarium head blight one of the most important Fusarium diseases? What other species are involved in the Fusarium disease complex and why is this an important factor in either treatment with NTP or mycotoxin production? Why are mycotoxins a high priority and what do they do? As important and severe as the authors describe mycotoxins as, there is no information as to why, apart from “severe health problems”. These are just three examples, though more depth and focus is required across the introduction and not just these three examples.
A whole paragraph is also dedicated to references for the use of NTP in medicine, the textile industry, forestry and food, but no description as to any of these uses. I think reference formatting has not been used appropriately here and this can be condensed. No further detail has been given about any of these industries apart from the fact they exist.
Methods, Results Discussion:
For a systematic review, MDPI Plants instructions for authors says, “Structured reviews and meta-analyses should use the same structure as research articles and ensure they conform to the PRISMA guidelines”. A checklist of the PRISMA guidelines can be found online.
I also think the authors need to perform a more comprehensive search of the literature to ensure they have found all relevant articles. I am not an expert in NTP, so some of the papers may not apply directly, but still could provide some context around what is currently occurring in this field. Examples include:
- Effects of Atmospheric-Pressure Cold Plasma Treatment on Deoxynivalenol Degradation, Quality Parameters, and Germination of Barley Grains. Feizollahi 2020.
- Use of Cold Atmospheric Plasma to Detoxify Hazelnuts from Aflatoxins. Siciliano, 2016
- Inhibitory effect of double atmospheric pressure argon cold plasma on spores and mycotoxin production of Aspergillus niger contaminating date palm fruits. Ouf, 2014
- Influence of Plasma Treatment in Open Air on Mycotoxin Content and Grain Nutriments. Kriz, 2015.
- Degradation of mycotoxins using microwave-induced argon plasma at atmospheric pressure. Park, 2007.
- A systematic study of the antimicrobial mechanisms of cold atmospheric-pressure plasma for water disinfection. Xu, 2020.
Results:
Fusarium inactivation (Table 1):
This section lacks any sort of cohesion and is merely just a listing of results from papers in the order of Table 1. This would be much more meaningful if the authors structure their manuscript in a way that highlights different results/experiments, rather than just a recounting of papers they have read.
The reviewed literature should be separated into sections based on some sort of criteria, such as host, or plasma treatment type, or observed results etc. That way meaningful links could be drawn between the reviewed literature. An example could be a section on “NTP effect on Fusarium spores” and a description of how NTP effects spores. The authors could then describe multiple facets of NTP usage on Fusarium and give the reader some structure in outcomes from the literature e.g. “NTP effect on Mycelium”, “NTP effect on host gene regulation”, “Effective of reactive species on Fusarium”.
Mycotoxin degradation (Table 2):
This section sees even less description than the previous. I would suggest the authors also incorporate my suggestion above about Fusarium inactivation, to the mycotoxin degradation discussion.
Discussion:
With the caveat that this might be a language issue or lack of experience with manuscript writing instead, I get the impression sometimes that the authors either did not have either the time, or the motivation to write this manuscript, as there have been multiple instances of lazy writing. An example is the following sentence “had the same outcome of deactivating Fusarium spp and other fungi and decontaminating the mycotoxins produced by these respective fungi in their respective study.”. The authors referring to “respective fungi and their respective study” gives me the feeling they are not giving me the information I need here to understand what they are saying. I find it better to explicitly say what you need to, don’t let your reader have to guess or go find previous information, as that ruins the flow of the reading. This is one example of this, but others can be found throughout the document.
There should also be an element of critical evaluation from the authors. What do all these papers mean, do they agree, or do they disagree? What are the missing areas of research? Where should this research be going?
Methods:
For a systematic review, this is not detailed enough. Please see my previous examples as well as check the PRISMA guidelines.
Specific issues
Abstract
L10-11: sentence should read “reduce maximum yield if left unchecked”
L11: “and are well documented”
L12: This entire sentence needs rewording
Introduction:
L22-24: There needs to be a reference to the APS website. What 108 plant species does this reflect? This number appears arbitrary. The Summerell reference needs to be properly formatted here too.
L29-30: What does the disease complex have to do with Fusarium’s treatment and mycotoxin production? If it’s important enough to be mentioned here, elaborate. Otherwise remove it.
L32-34: This reads as mycotoxins infect crops. Do you mean pathogenic fungi infect crops and produce mycotoxins? The grammar in this sentence needs fixing.
L34-35: Little information is presented here about mycotoxins and why the WHO has declared them a high priority.
L57: Tell the reader a bit about NTP devices, what in their differences make them more/less applicable to Fusarium inactivation and mycotoxin degradation? Do they produce different forms/amounts of reactive species? Are any able to be used in field to process plants and infections in an agricultural setting?
Results:
L107: Keep formatting of units consistent throughout whole document. “3 s” vs “3s”.
Discussion:
L169: “It is essential to understand the different underlying mechanisms of fungal inactivation caused by non-thermal plasma”. However I don’t believe the authors have given us enough information on what these underlying mechanisms are. They mention structural changes, reaction with active species and that oxidation of amino acids and nucleic acids are detrimental for example, but no further details about what structural changes occur, what reactive species are formed or how oxidation of amino acids and nucleic acids affect the pathogen.
L194-197: This would be a good point for the authors to comment on mycotoxin degradation. For example, if one hypothesis is that mycotoxins are being degraded at temperatures higher than 60 degrees, could this potentially affect food products? Are there more accessible methods for achieving this temperature than NTP? Is cooking of food alone good enough to degrade mycotoxins? Iqdiam and Hamad say that different mycotoxins have different chemical structures and go through a multistep degradation process. Is NTP better at dealing with specific mycotoxins?
L208: This sentence needs to be fixed grammatically and a better description of why it is like “killing two birds with one stone”. The sentence seems incomplete here without at least a little description as to the analogy.
References:
L300: Correct the formatting of H2O2
Final thoughts:
The comments above are genuinely trying to help improve the document and I hope the authors won’t be too disheartened by them. I think as it stands there is a good skeleton here to base a comprehensive review from, but the document will need some significant reworking in terms of the style, structure and content. It may be worth considering resubmitting as a review, rather than a systematic review if there is difficulty matching the PRISMA guidelines. In addition, a good English review of the document is required as there are several grammatical issues. I wish the authors good luck with improving this manuscript.

Author Response
Dear Reviewer,
We thank you for your patience and for providing critical review for our manuscript to make it better. Please find the answers (in italics) to your comments/suggestions below:
Overall:
This review by Drs Doshi and Šerá investigates previous literature on the effect of Non-thermal plasma and it’s potential usage in inactivation of Fusarium species for Agriculture and mycotoxin degradation for the food industry. The use of NTP in agriculture poses an interesting replacement for pesticides and the aim of the review article appears to address it’s uses and previous findings associated with experimentation on Fusarium and a handful of other fungal species.
I find the submitted review article to be lacking in its ability to bring together a cohesive overview of the technology and previous findings in the literature. The manner in which the literature is presented is superficial and presented in a way that does not emphasise how the various findings are linked and is instead presented in an almost dot-point like fashion, ignoring any meaningful attempt at reviewing the topic as a whole. While it appears the authors have formatted their manuscript to be a systematic review, the PRISMA guidelines have not been followed.
The main articles reviewed are found in two tables in the submitted manuscript and the text associated with these tables addresses each of these articles one after the other, with little attempt at finding patterns within the literature and linking them together. I understand that the authors have tried to write a systematic review, however, this could be much better presented as a series of subjects important in the review (elaborated later on in my comments) where the authors discuss these subjects and the associated literature, rather than just listing papers. Good examples of systematic review structure can be found in https://www.mdpi.com/2223-7747/12/2/305 and https://www.mdpi.com/2223-7747/12/2/261
I also find detail lacking in the abstract and introduction, where thoughts appear to be unfinished and the reader is often left to their own devices to figure out what the author is trying to say.
In addition, the entire document needs to be edited for grammar.
I have included examples to all these concerns below. The examples however are not exhaustive as a rework of the entire document is required.
We would like to thank you very much for your in-depth critical review. We have corrected the manuscript as per your comments and suggestions.
Comments per section:
Abstract:
The abstract gives no indication to the reader about the outcomes of the review. Does NTP work to inactivate Fusarium mycotoxin? In fact, mycotoxins are not mentioned at all in the abstract. This abstract is too superficial and short to give the reader any indication about what the review of the literature found. I found myself unsure about what the authors were trying to review based on the abstract alone. For a systematic review, be sure to follow the PRISMA for Abstracts checklist.
The abstract has been modified accordingly
Introduction:
The introduction comes across very shallow and needs to be improved upon. Many points are raised and not elaborated on, examples include “one of the most important is the Fusarium head blight”, “The diseases caused by Fusarium spp. often involves more than one species in a form of a complex” and “World Health Organization (WHO) and Food and Agriculture Organization (FAO) has declared a high priority on mycotoxins”.
The information has been edited.
Why is Fusarium head blight one of the most important Fusarium diseases? What other species are involved in the Fusarium disease complex and why is this an important factor in either treatment with NTP or mycotoxin production? Why are mycotoxins a high priority and what do they do? As important and severe as the authors describe mycotoxins as, there is no information as to why, apart from “severe health problems”. These are just three examples, though more depth and focus is required across the introduction and not just these three examples.
The information has been added and edited accordingly
A whole paragraph is also dedicated to references for the use of NTP in medicine, the textile industry, forestry and food, but no description as to any of these uses. I think reference formatting has not been used appropriately here and this can be condensed. No further detail has been given about any of these industries apart from the fact they exist.
The reason for no description is because we wanted to show the different applications of NTP in different sectors and how it is also gaining attention in agriculture. Detailed description of them is irrelevant here and our focus is particularly on Fusarium spp inactivation and mycotoxin degradation.
Methods, Results Discussion:
For a systematic review, MDPI Plants instructions for authors says, “Structured reviews and meta-analyses should use the same structure as research articles and ensure they conform to the PRISMA guidelines”. A checklist of the PRISMA guidelines can be found online.
Changes have been performed using PRISMA guidelines and edited accordingly
I also think the authors need to perform a more comprehensive search of the literature to ensure they have found all relevant articles. I am not an expert in NTP, so some of the papers may not apply directly, but still could provide some context around what is currently occurring in this field. Examples include:
- Effects of Atmospheric-Pressure Cold Plasma Treatment on Deoxynivalenol Degradation, Quality Parameters, and Germination of Barley Grains. Feizollahi 2020.
- Use of Cold Atmospheric Plasma to Detoxify Hazelnuts from Aflatoxins. Siciliano, 2016
- Inhibitory effect of double atmospheric pressure argon cold plasma on spores and mycotoxin production of Aspergillus niger contaminating date palm fruits. Ouf, 2014
- Influence of Plasma Treatment in Open Air on Mycotoxin Content and Grain Nutriments. Kriz, 2015.
- Degradation of mycotoxins using microwave-induced argon plasma at atmospheric pressure. Park, 2007.
- A systematic study of the antimicrobial mechanisms of cold atmospheric-pressure plasma for water disinfection. Xu, 2020.
Thank you for the articles. We used the articles that were relevant in the text. Since we used, Web of Science Core collection, we wished to keep the search very focus on the two topics i.e a) Fusarium inactivation and b) mycotoxin degradation
Results:
Fusarium inactivation (Table 1):
This section lacks any sort of cohesion and is merely just a listing of results from papers in the order of Table 1. This would be much more meaningful if the authors structure their manuscript in a way that highlights different results/experiments, rather than just a recounting of papers they have read.
The reviewed literature should be separated into sections based on some sort of criteria, such as host, or plasma treatment type, or observed results etc. That way meaningful links could be drawn between the reviewed literature. An example could be a section on “NTP effect on Fusarium spores” and a description of how NTP effects spores. The authors could then describe multiple facets of NTP usage on Fusarium and give the reader some structure in outcomes from the literature e.g. “NTP effect on Mycelium”, “NTP effect on host gene regulation”, “Effective of reactive species on Fusarium”.
The changes have been performed
Mycotoxin degradation (Table 2):
This section sees even less description than the previous. I would suggest the authors also incorporate my suggestion above about Fusarium inactivation, to the mycotoxin degradation discussion.
The changes have been performed
Discussion:
With the caveat that this might be a language issue or lack of experience with manuscript writing instead, I get the impression sometimes that the authors either did not have either the time, or the motivation to write this manuscript, as there have been multiple instances of lazy writing. An example is the following sentence “had the same outcome of deactivating Fusarium spp and other fungi and decontaminating the mycotoxins produced by these respective fungi in their respective study.”. The authors referring to “respective fungi and their respective study” gives me the feeling they are not giving me the information I need here to understand what they are saying. I find it better to explicitly say what you need to, don’t let your reader have to guess or go find previous information, as that ruins the flow of the reading. This is one example of this, but others can be found throughout the document.
There should also be an element of critical evaluation from the authors. What do all these papers mean, do they agree, or do they disagree? What are the missing areas of research? Where should this research be going?
The information is added in the text.
Methods:
For a systematic review, this is not detailed enough. Please see my previous examples as well as check the PRISMA guidelines.
The changes are performed in the revised version.
Specific issues
Abstract
L10-11: sentence should read “reduce maximum yield if left unchecked”
Changes performed
L11: “and are well documented”
Changes performed
L12: This entire sentence needs rewording
Changes performed
Introduction:
L22-24: There needs to be a reference to the APS website. What 108 plant species does this reflect? This number appears arbitrary. The Summerell reference needs to be properly formatted here too.
The changes have been made in the text.
L29-30: What does the disease complex have to do with Fusarium’s treatment and mycotoxin production? If it’s important enough to be mentioned here, elaborate. Otherwise remove it.
The sentence has been modified
L32-34: This reads as mycotoxins infect crops. Do you mean pathogenic fungi infect crops and produce mycotoxins? The grammar in this sentence needs fixing.
The sentence has been modified
L34-35: Little information is presented here about mycotoxins and why the WHO has declared them a high priority.
The information is added.
L57: Tell the reader a bit about NTP devices, what in their differences make them more/less applicable to Fusarium inactivation and mycotoxin degradation? Do they produce different forms/amounts of reactive species? Are any able to be used in field to process plants and infections in an agricultural setting?
The information is added in the text.
Results:
L107: Keep formatting of units consistent throughout whole document. “3 s” vs “3s”.
Correction done everywhere
Discussion:
L169: “It is essential to understand the different underlying mechanisms of fungal inactivation caused by non-thermal plasma”. However I don’t believe the authors have given us enough information on what these underlying mechanisms are. They mention structural changes, reaction with active species and that oxidation of amino acids and nucleic acids are detrimental for example, but no further details about what structural changes occur, what reactive species are formed or how oxidation of amino acids and nucleic acids affect the pathogen.
Thank you for the comment. We have included the details in the text.
L194-197: This would be a good point for the authors to comment on mycotoxin degradation. For example, if one hypothesis is that mycotoxins are being degraded at temperatures higher than 60 degrees, could this potentially affect food products? Are there more accessible methods for achieving this temperature than NTP? Is cooking of food alone good enough to degrade mycotoxins? Iqdiam and Hamad say that different mycotoxins have different chemical structures and go through a multistep degradation process. Is NTP better at dealing with specific mycotoxins?
We apologize for the incorrect interpretation. We have made the correction and the changes accordingly.
L208: This sentence needs to be fixed grammatically and a better description of why it is like “killing two birds with one stone”. The sentence seems incomplete here without at least a little description as to the analogy.
It has been removed from the text.
References:
L300: Correct the formatting of H2O2
Correction performed
Final thoughts:
The comments above are genuinely trying to help improve the document and I hope the authors won’t be too disheartened by them. I think as it stands there is a good skeleton here to base a comprehensive review from, but the document will need some significant reworking in terms of the style, structure and content. It may be worth considering resubmitting as a review, rather than a systematic review if there is difficulty matching the PRISMA guidelines. In addition, a good English review of the document is required as there are several grammatical issues. I wish the authors good luck with improving this manuscript.
Thank you for your critical review, comments and suggestions. We have revised the manuscript as per your comments, suggestions.
Reviewer 3 Report
The article is well written and provides published literature on cold plasma effects on an important pathogenic fungus, Fusarium spp.
Author Response
Dear Reviewer,
We thank you for your patience and for providing us your valuable comment.
Reviewer 4 Report
Manuscript ID: plants-2158216. Type of manuscript: Review. Title: Role of Non-thermal plasma in Fusarium inactivation and mycotoxin decontamination. Authors: Pratik Doshi *, Božena Šerá
The authors described a literature search on plasma treatment of Fusarium spp.
Comments:
Page 2, Line 50.
“Physical plasma is the fourth state of matter and is defined by Domonkos et al. [8] as partially or fully ionized quasi-neutral substance that is made up of electrons, ions, neutral particles, molecules in the ground or excited state, radical species and quanta of electromagnetic radiation (UV photons and visible light).”
The correct definition of plasma can be found in any textbook, and it does not belong to Domonkos et al. This definition in the manuscript is incorrect. Plasma is not a single substance and consists from many components, such as electrons, ions, neutral particles, molecules in the ground or excited state, radicals and quanta of electromagnetic radiation.
Page 3, Table 1 is uninformative and does not contain information about the result of the plasma treatment.
Pages 5 – 6. Information on reproducibility of results should be added. Many authors recommended completely different times of the procedures.
Page 6, Table 2 is uninformative and does not contain information about the result of plasma treatment.
Page 9. Fig. 2 is not a Figure, but a simple diagram. I do not see in the present manuscript any new proposed mechanisms or, at least, a working hypothesis about the inactivation of Fusarium and degradation of mycotoxins.
In my opinion, this paper needs a significant revision. What is new in this manuscript?
Author Response
Dear Reviewer,
We thank you for your patience and for providing critical review for our manuscript to make it better. Please find the answers (in italics) to your comments/suggestions below:
Reviewer 2:
Page 2, Line 50.
“Physical plasma is the fourth state of matter and is defined by Domonkos et al. [8] as partially or fully ionized quasi-neutral substance that is made up of electrons, ions, neutral particles, molecules in the ground or excited state, radical species and quanta of electromagnetic radiation (UV photons and visible light).”
The correct definition of plasma can be found in any textbook, and it does not belong to Domonkos et al. This definition in the manuscript is incorrect. Plasma is not a single substance and consists from many components, such as electrons, ions, neutral particles, molecules in the ground or excited state, radicals and quanta of electromagnetic radiation.
Thank you for the comment. We have made the correction in the text. Please refer to Lines 78-81
Page 3, Table 1 is uninformative and does not contain information about the result of the plasma treatment.
The information is added in the table.
Pages 5 – 6. Information on reproducibility of results should be added. Many authors recommended completely different times of the procedures.
Regarding the reproducibility of the results, the information is already mention in the Discussion section 3.1. We believe that rather than the apparatus per se, the actual working conditions such as gas, voltage, power, treatment time etc, are the most important parameters that needs to be considered. Please refer to Lines 257-261.
Page 6, Table 2 is uninformative and does not contain information about the result of plasma treatment.
The information is added in the table
Page 9. Fig. 2 is not a Figure, but a simple diagram. I do not see in the present manuscript any new proposed mechanisms or, at least, a working hypothesis about the inactivation of Fusarium and degradation of mycotoxins.
The diagram is removed entirely.
In my opinion, this paper needs a significant revision. What is new in this manuscript?
Thank you very much for your comment. NTP is a fairly new technology in agriculture unlike the other sectors such as textile, medicine etc. In this review, we focused only on the Fusarium spp and mycotoxins produced by them and NTP effects on them. Although there are not quite as much of conclusive literature available, this review aimed to summarize the literature and produce a synthesis. We have also proposed a novel IPM strategy which could possibly be considered in the future, given the fact that both the treatments i.e NTP and biological control are sustainable, chemical-free, environmental friendly; after a thorough research before it is put onto the field trials.
Round 2
Reviewer 1 Report
The authors have modified the manuscript accordingly nad comprehensively.
Author Response
Dear Reviewer, Thank you very much for your review. We appreciate your time and effort in reviewing our manuscript. We have edited the English language and style and also minor spell check it performed.
Reviewer 2 Report
Overall comments
I would like to first start by congratulating the authors for turning around their changes in such a short time period and for making a thorough attempt at addressing the issues. The resubmitted manuscript is significantly improved. I think the manuscript would still benefit from an English editor as there are a few minor grammatical issues.
Abstract:
The abstract is improved and now reflects better what is included within the manuscript. No changes required.
Introduction:
I feel the introduction has vastly improved and I commend the authors on their edits. Minor comments below.
L32: This is still out of context, 83 of 108 plant species. What are these 108 plant species and why are there only 108? There are many more than 108 agriculturally important plant species. Unless you can find a reference that has an indication of how many total host species there might be, I think you could replace this with some specifics. Instead E.g. Table 1 in https://www.horticulture.com.au/globalassets/hort-innovation/resource-assets/ny11001--fusarium.pdf is a good example. Hopefully this helps.
L35: Acronyms shouldn’t be used to start a sentence. This should also continue from the last paragraph, not be a new paragraph.
L36-37: “Buerstmayr et al. [5] stated in their review that Nganje et al. in 2004”. Don’t use a reference to reference a reference. Just say “Nganje et al. in 2004, estimated the loss of approximately US$ 2.5 billion attributable to FHB on wheat and barley for the period 1993–2001”. No need to include Buerstmayr here.
L87-92: Great! Thanks for the extra information here.
Results:
Table 1 and 2: I appreciate the ‘salient results’ sections of the tables. However, I’m honestly not sure if this is better in the text or in the table. I will leave this up to the author’s decision. My uncertainty here is that the text is a little difficult to read in a thin column. But I think it is useful having a summary here.
Discussion:
L457: “Nevertheless, there have been other studies that also show the success of NTP.” This sentence can probably be removed, it’s a little awkward and doesn’t add anything that the previous sentence hasn’t said.
The second paragraph of the discussion “The unique quality of NTP …” Might be a better paragraph to start your discussion with. The first paragraph in the discussion is more results than discussion. You want to have the most important findings explained first in your discussion. Again, I will leave this to the author’s discretion, my recommendation would be to really drive home the major finding straight away and follow that up with your evidence. The wording I think needs a little reworking in the second paragraph. However, what I’m getting from this though is that while NTP can use different combinations of gasses and apparatus, that the same outcome of inactivating fungi is found irrespective. You also mention this in the conclusions, so I think highlighting this in the first paragraph would really make this clear.
For the first paragraph of section 3.2 (L571), you do this, straight away I know NTP can degrade mycotoxins and especially DON. This reads well.
Materials and methods:
My only comment here is about restricting your search to Web of Science. This might have been an access limitation for your institution. It might be worth mentioning these limitations and what you did to mitigate them.
Final thoughts:
The authors have made significant improvements to their document and as such I believe the document is now more comprehensive and much better highlights the research in the usage of NTP against Fusarium and Mycotoxins. With a final pass of English editing, this manuscript should be suitable for publication.
Author Response
Dear Reviewer, The responses to your comments/suggestions are in italics below.
--------------------------------------------------------------
Overall Comments
I would like to first start by congratulating the authors for turning around their changes in such a short time period and for making a thorough attempt at addressing the issues. The resubmitted manuscript is significantly improved. I think the manuscript would still benefit from an English editor as there are a few minor grammatical issues.
Thank you very much for your comments. We have edited the English language and style and also minor spell check it performed
Abstract:
The abstract is improved and now reflects better what is included within the manuscript. No changes required.
Introduction:
I feel the introduction has vastly improved and I commend the authors on their edits. Minor comments below.
L32: This is still out of context, 83 of 108 plant species. What are these 108 plant species and why are there only 108? There are many more than 108 agriculturally important plant species. Unless you can find a reference that has an indication of how many total host species there might be, I think you could replace this with some specifics. Instead E.g. Table 1 in https://www.horticulture.com.au/globalassets/hort-innovation/resource-assets/ny11001--fusarium.pdf is a good example. Hopefully this helps.
The whole sentence is now modified. Thank you for your article.
L35: Acronyms shouldn’t be used to start a sentence. This should also continue from the last paragraph, not be a new paragraph.
Acronyms have been replaced with their full forms throughout the sentences.
L36-37: “Buerstmayr et al. [5] stated in their review that Nganje et al. in 2004”. Don’t use a reference to reference a reference. Just say “Nganje et al. in 2004, estimated the loss of approximately US$ 2.5 billion attributable to FHB on wheat and barley for the period 1993–2001”. No need to include Buerstmayr here.
We edited the sentence.
L87-92: Great! Thanks for the extra information here.
You are most welcome.
Results:
Table 1 and 2: I appreciate the ‘salient results’ sections of the tables. However, I’m honestly not sure if this is better in the text or in the table. I will leave this up to the author’s decision. My uncertainty here is that the text is a little difficult to read in a thin column. But I think it is useful having a summary here.
We prefer to keep the salient results for the readers to understand the summary as you rightly mentioned.
Discussion:
L457: “Nevertheless, there have been other studies that also show the success of NTP.” This sentence can probably be removed, it’s a little awkward and doesn’t add anything that the previous sentence hasn’t said.
Removed.
The second paragraph of the discussion “The unique quality of NTP …” Might be a better paragraph to start your discussion with. The first paragraph in the discussion is more results than discussion. You want to have the most important findings explained first in your discussion. Again, I will leave this to the author’s discretion, my recommendation would be to really drive home the major finding straight away and follow that up with your evidence. The wording I think needs a little reworking in the second paragraph. However, what I’m getting from this though is that while NTP can use different combinations of gasses and apparatus, that the same outcome of inactivating fungi is found irrespective. You also mention this in the conclusions, so I think highlighting this in the first paragraph would really make this clear.
We have edited it as per your comments.
For the first paragraph of section 3.2 (L571), you do this, straight away I know NTP can degrade mycotoxins and especially DON. This reads well.
Sentence edited as per your comments.
Materials and methods:
My only comment here is about restricting your search to Web of Science. This might have been an access limitation for your institution. It might be worth mentioning these limitations and what you did to mitigate them.
We have justified for restricting the search to core collection.
Final thoughts:
The authors have made significant improvements to their document and as such I believe the document is now more comprehensive and much better highlights the research in the usage of NTP against Fusarium and Mycotoxins. With a final pass of English editing, this manuscript should be suitable for publication.
Dear Reviewer, We appreciate your time and effort in reviewing our manuscript. We have edited the English language and style and also minor spell check it performed
Reviewer 4 Report
The revised manuscript can be accepted for publication
Author Response
Dear Reviewer,
Thank you for your comments. We appreciate your time and effort in reviewing our manuscript. We have edited the English language and style and also minor spell check it performed.